

# Brief Communication : Monitoring active layer dynamic using a lightweight nimble Ground-Penetrating Radar system. A laboratory analog test case.

Emmanuel Léger[1], Albane Saintenoy[1], Mohammed Serhir[3], François Costard[1], and Christophe Grenier[2,†]

[1]Université Paris-Saclay, CNRS, GEOPS, 91405, Orsay, France

[2,†]Laboratoire des Sciences du Climat et de l'Environnement, Université Paris-Saclay,IPSL/LSCE, UMR 8212 CNRS-CEA-UVSQ,Orme des Merisiers, Gif-sur-Yvette Cedex,France. Deceased

[3]Université Paris-Saclay, CentraleSupélec, CNRS, Laboratoire de Génie Electrique et Electronique de Paris, 91192, Gif-sur-Yvette, France

**Correspondence:** emmanuel.leger@universite-paris-saclay.fr

**Abstract.** Monitoring active layer dynamic is critical for improving the near surface thermal and hydrological process understanding. This study presents the laboratory test of a low-cost Monitoring Ground-Penetrating Radar (GPR) system within a laboratory experiment of active layer freezing and thawing monitoring. The system is a in-house built low power monostatic GPR antenna coupled with a reflectometer piloted by a single board computer, tested prior to field deployment. The corre-
spondence between the frozen front electromagnetic reflection and temperature allowed the better understanding of the frozen front/bottom of the active layer reflection and the intrinsic permittivity of the frozen layer.

## 1 Introduction

The freezing and thawing cycles of frozen material in cryosphere environment are important processes not only for water and temperature redistribution, but are critical for nutrient cycles (Nitrogen and Carbon) and atmosphere gas release. As such, mon-
itoring, non destructively, the seasonal soil freezing and thawing in high latitude contexts has been an extensive topic of interest in cold regions advances. Previous researchers have illustrated the suitability of geophysical methods, especially Ground Penetrating Radar (GPR) techniques, for monitoring active layer freezing and thawing (Liu and Arcone, 2003; Van der Kruk et al., 2009; Jadoon et al., 2015; Van Der Kruk et al., 2007; Westermann et al., 2010; Steelman and Endres, 2009; Sudakova et al., 2021). Reflection from the permafrost table, or active layer base, were compared in GPR profiles with acquisitions repeated at
different times (Westermann et al., 2010; Sudakova et al., 2021) The first 10 cm soil freezing/thawing cycles over 9 days was monitored using a static antenna set of an air-launched off-ground GPR system (Jadoon et al., 2015). With the same steady monitoring approach, common mid-point techniques were used to study electromagnetic wave dispersion and frequency to infer active layer freezing and thawing acting as waveguide (Van der Kruk et al., 2009; Steelman and Endres, 2009). As far as we know, no studies were performed using monostatic on-ground antenna monitoring on the same location through time apart
a first attempt in laboratory with a commercial antenna system (Saintenoy et al., 2005). Here we present a novel combination of low-cost/-energy nimble GPR mono-static ground-coupled antenna with a reflectometer in conjunction with a small array of





thermal and volumetric water content sensors for monitoring active layer freezing and thawing during laboratory experiments. The study is thought as a first test case on a active layer laboratory analog before near future field deployments.

## 2 Materials and Methods

### 2.1 Laboratory Modeled Active Layer characteristics

The laboratory-simulated active layer was made of Fontainebleau fine sand ($D_{50}$= 200 $\mu/m$) contained in a cylindrical reservoir of 52 cm height and a 84-90 cm diameter, vase shaped as depicted and drafted in Figures 1-a) and 1-b). At the base of the column a system of copper pipes was installed and connected to an external cryostat flowing glycol in the copper pipes at -10°C (see additional materials). The freezing copper pipes were surrounded by a gravel layer, used as drainage horizon and spreading the weight of the above sand layer. A thin layer of aluminum fold with multiple holes was placed above the gravel layer to create a strong bottom reflector. A thin geo-textile sheet was set on top of the drainage-cold layer for preventing the sand to be flushed out and to keep minimal the frost heave according to Henry (1990). The inner-diameter of the cylinder was covered by polystyrene foam blankets all around its inner diameter, to ensure as much as possible radial thermal insulation and keep only the top (atmosphere) and bottom (copper-cryostat pipe) boundaries (-10°C) as open flow conditions. The cylindrical tank was hydraulically connected to tap water with a buffer reservoir system allowing to set water table at different heights. This system allowed us to properly and incrementally saturate the sand material while compacting it in the reservoir. The sand hydrodynamic parameters, such as the Mualem (1976) van Genuchten (1980)'s parameters are presented in supplementary materials and can be found in Léger et al. (2020). A well-proven technique was used to create a homogeneous porous medium of more than 400 kg of saturated soil (Costard et al., 2021) : we manually compacted layers of 10 cm each by maintaining the water level higher than the considered layer and then put them on top of each other until 0.47 m thickness was reached. The sand was packed to a porosity estimated to be 0.39. There might be some small interfaces (thin unconformities), but they remain insignificant (we verified post experimentally when excavating the sand). The low temperature external atmospheric conditions were set using the cold room facility at the GEOPS laboratory (University Paris Saclay, France) dedicated to the physical modeling in periglacial geomorphology (e.g. Costard et al. (2021)). The external temperature of the cold room was maintained slightly above 1°C, throughout the freezing phase. For shortening the thawing time length, the thawing phase was performed by shutting down all freezing processes, open the cold room, and equilibrate with ambient temperature (20°C). As drawn in Figure 1-a), a string of 10 thermistors (PT100 with ± 0.1 °C) were set from the bottom of the sand layer to the surface at 0, 2.5, 5, 7.5, 10, 15, 20, 30, 40 and 47 cm centimeter from the bottom. Complementing these thermal measurements, 3 volumetric water content sensors (Decagon Terros 12) were set diametrically opposed to the thermistors string at 0, 10, 20 and 30 cm from the bottom of the sand layer. In addition to the sensor network, a handheld thermal camera (Flir TG297) was used to check the thermal isolation continuity of the cylindrical reservoir, notably the thermal insulating polystyrene layer on the curvilinear sides of the reservoir.



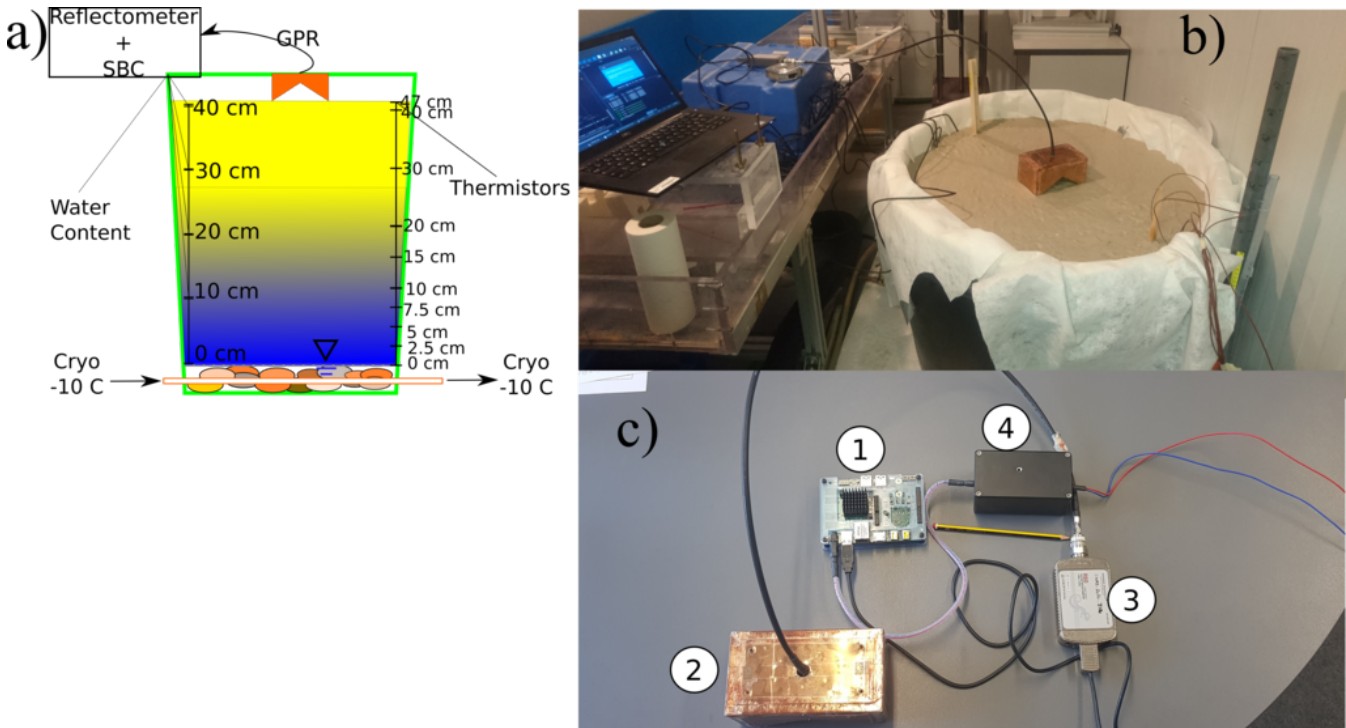

**Figure 1.** Experimental setup : a) draft of the experiment, b) picture of the GPR prototype with the antenna and the reflectometer in the cold room, c) picture of the 4 main components of the M-GPR system: Udoo SBC computer (1), Butterfly antenna(2), reflectometer(3) and (4)power-supply timer plugged to a 12 V car battery

## 2.2 Autonomous Monitoring Ground Penetrating Radar System : M-GPR

The M-GPR system comprises an ultra-wide band antenna designed in the GeePs Laboratory operating in frequency band
1 GHz – 6 GHz. The antenna is a fractal-folded complementary bowtie antenna (Liu et al., 2018). It is combined with the R60 Copper Mountain USB vector network analyzer to measure the reflection coefficient (in this case $S_11$) of antenna placed in front of the area under test (See Figure 1-b) and c) ). The antenna design has been engineered with fractals in order to enlarge the covered bandwidth. It is enclosed in a shielded cavity to reduce backward radiation and to protect from external electromagnetic perturbations. Inside the cavity, absorbers are placed to prevent the antenna performance degradation while
reducing multiple reflections inside the cavity. Finally, the antenna is 30° bended in order to focus the energy in the investigated area direction.

The R60 Copper Mountain reflectometer is controlled (via serial USB connection) using an UDOO single board computer. The board is powered using a stabilized current source (12V) and a timer (CRONTAB) to allow power/sleep modes for long time monitoring campaign. Indeed, for the presented results, a stack of 10 traces was performed regularly in time (every 5
minutes) where the real and imaginary part of $S_{11}$ are recorder over 801 points covering the frequency band 0.2 GHz – 5 GHz.



The frequency domain measurement data are inverse Fourier transformed to generate time domain radargrams with a time window of 30 ns.

The expected dynamic range of measured data can reach up to $90\ dB$ using the R60 Copper Mountain reflectometer and adopting the step frequency technique. In addition, on a $12\ V\ 60Ah$ battery, the GPR system offers the possibility to monitor
during a month with a measurement every 5 minutes including the pre-heating of the reflectometer and a stack of 10 traces. Using the M-GPR system, one can configure the timer (CRONTAB) to wake-up the UDOO board computer every 12 hours in order to monitor the area under investigation during months.

## 2.3 Electromagnetic Wave Velocity As A Function Of Soil Dielectric Properties

The mono-static antenna emits and receives frequency sweep of spherical electromagnetic (EM) waves in response to the
reflectometer excitation at different frequencies. The receiving antenna converts the incoming EM fields to electrical signals. Following the work of Annan (1999), the velocity of an electromagnetic wave propagating in a non-magnetic soil, with low electrical conductivity, can be approximated by :

$$v = \frac{c}{\sqrt{\varepsilon'}}\ , \tag{1}$$

where $\varepsilon'$ denotes the real part of the relative dielectric permittivity and $c = 0.3$ m/ns is the velocity of EM waves in air.
The dielectric permittivity depends on the media component content, such as water, air, minerals and ice content. We used the volumetric mean Complex Refractive Index Model (CRIM) (Birchak et al., 1974), where the relative dielectric permittivity of the porous geological material, $\varepsilon_b$, is a function of the material porosity, its state of saturation and the respective permittivity of each of its individual constituents. For the thawed water-saturated case, the Fontainebleau sand medium is bi-phasic composed of water and silica :

$$\varepsilon_b^\gamma = \phi\varepsilon_w^\gamma + (1-\phi)\varepsilon_s^\gamma\ , \tag{2}$$

where $\varepsilon_w$ and $\varepsilon_s$ are respectively the relative dielectric permittivity of water and of silica, $\phi$ is the porosity, and $\gamma$ is an empirical coefficient that depends on soil structure. In this study, we set $\varepsilon_w = 81.1$, $\varepsilon_s = 4.2$, $\phi = 0.39$ and $\gamma = 0.5$ as in (Léger et al., 2020). We assumed all media were non conductive and no dispersion effect was taken into account. The frozen case was similar to eq. 2 except that the liquid water phase was replaced by its solid phase,

$$\varepsilon_b^\gamma = \phi\varepsilon_i^\gamma + (1-\phi)\varepsilon_s^\gamma\ , \tag{3}$$

where $\varepsilon_w$ and $\theta_i$ are the relative dielectric permittivity of ice (set here to 3.1) and the volumetric ice content respectively.

## 2.4 Electromagnetic modeling

For simulating our M-GPR data we used the open source software GprMax2D (Warren et al., 2016), in two dimensions, as presented in Léger et al. (2020). The EM model was set to 1.2 m wide and 0.6 m high, larger than the real model to avoid
side effects, filled up to 0.48 m with saturated sand following the permittivity distribution presented in section 2.3. The cell



size were 5x5 mm$^2$, a Perfect Electrical Conductor (PEC) layer was set at the bottom of the sand layer and the model domain sides were padded with 10 cell thick absorbing boundaries. The GPR antennae was modeled by a Hertzian dipole polarized in Z direction (perpendicular to the 2D plane model), emitting a Ricker waveform centered on 1.5 GHz.

## 3 Results

### 3.1 Temperature and volumetric liquid water content time series

The temperature time series measured during one of the experiments are presented in Figure 2. During the equilibrating phase the surface temperature sensor drops the fastest to the constraint temperature (+1°C). The thermal time series exhibit little 8-hour cycles for the shallowest sensor at the surface due to the chamber cryostat defrost cycles, maintaining the cold room at +1 °C. The bottom freezing phase start is symbolized by the blue dashed vertical line at. The temperature reaches within few hours the zero isotherm with a typical curtain effect for the sensors close to the cold system (z = 0, 2.5, 5 cm), while almost a day is required for sensors from 20 cm to the top. Near surface and surface sensors (40 cm and Surface) are within the 0°C. Once the 0°C isotherm is crossed, the temperature are following an exponential decrease as expected (e.g. Hinzman et al. (1998)). The depth-temperature profile derived from the time series presented in 2-c) exhibits the depth of the zero-isotherm reaching the ground surface after 125 h from the start of the experiment. Once the freezing phase done, the thawing phase exhibits a fast thawing for the first 30 cm below the surface while it takes more time deeper, with an optimum at 10 cm corresponding to the balance between heat dissipation and proximity to the bottom freezing cold source. The heat dispersion is as well visible on the different evolution of the curve with depth.

The volumetric liquid water content on the column is homogeneous during the equilibrating phase, Figure 2-b) except the 20-cm sensor being a bit lower comparing to the rest of the column, probably due to a mis-calibration or water-unsaturated pores nearby. During the freezing phase a little bump is observed right before dropping because of water freezing. This bump, wider as the sensor height is increasing, corresponds to liquid water pushed up by the frozen front. The 30-cm sensor is the last to be frozen and the first to be thawed.

Soil Freezing Characteristics (SFCs) at different depths were determined from volumetric water content and temperature data collected during the experiment (Figure 2-d). No supercooling effect was observed as those observed in more complex soils (Ren and Vanapalli, 2020), neither a strong zero curtain effect. Hysteresis in the SFCs are observed for each depths due to air and/or water and/or ice entrapment.

### 3.2 Ground Penetrating Radar Monitoring system (M-GPR)

The simulated radargram is presented in Figure 3-a). Each trace has been computed for a dielectric permittivity distribution (derived from Equation 3) corresponding to a frozen layer increasing from 0.02 m up to 0.3 m with 2 cm step. Six of them are presented Figure 3-(b) The two-way travel-time of the reflection on the PEC layer (labelled A) has been computed for



each permittivity distribution (green dots on Figure 3-(a)). Two reflections (B) and (C) are arriving earlier then reflection (A), directly linked to the frozen/thawed water saturated sand transition.

In the experimental M-GPR radargram, Figure 3-(c), three main reflections are present, labelled A, B and C. Comparing the experimental radargram to the simulated one, we find visual correspondences strengthening our assumption on the first C reflection corresponding to the frost table while the B reflection corresponds to the bottom of the transition zone, and A to the aluminum sheet. As opposed to the numerical case, B and C reflections are difficult to discriminate during the freezing phase, easier to distinguish during the thawing phase. The dynamic used in the numerical modeling (keeping a fixed transition zone thickness) does not fit the experimental data (where the transition thickness is clearly increasing with experiment time in Figure 2-(c)). Beyond the reflections similitude, the amplitude and wavelet are different, mostly due to the loss not taken into account in the modeling, while present in the experimental radargram.

These B and C reflections give us information on the thickness, the homogeneity and the dynamic of the active layer during the freezing and the thawing phases. We especially detect the speed of thawing being faster that the speed of freezing, and water redistribution in the near surface.

Substracting the TWTs of these reflections ($\text{TWT}_A$-$\text{TWT}_C$) and comparing it with the $0°$ isotherm height inferred from the thermistor time series, gives a linear relationship inversely proportional to the squared root of the bulk permittivity. By adjusting the linear regression with Theil Sen robust regression at 90% confidence interval we retrieve a bulk permittivity at $\varepsilon = 4.54 \pm 0.1$, similar to the literature for saturated frozen sand (e.g. Arcone et al. (1998)).

## 4  Discussions, future works and conclusions

The M-GPR system has been capable to deliver the evolution of the thaw depth and the average soil water content between the surface and the freeze-thaw interface for the freezing and the thawing period. For the laboratory conditions encountered in the study, the efficiency of M-GPR method is estimated to be less than a centimeter, being sufficient to secure spatial differences in the thaw depth very small dynamic. The main factors limiting the accuracy in the present study are: (1) the frost heave through the experiment and, (2) the surface condition changing the antenna impedance. The latter could be accounted for by using a bi-static antenna, allowing to retrieve the near surface hydric conditions and thus the impedance of the antenna. The first issue regarding the frost heave is easy to track in a laboratory experiment, here only 1-2 cm heave were observed since we used geotextile, while it is harder in the case of field studies. Beyond frost heave, soil tilting and surface de-planarization can happen and lead to loss of verticality and miscalculate the depth of the frost table. The M-GPR system by its agility will be of used for better understanding the GPR reflection in the case of supra-permafrost water layer, especially if the reflection is due to the unsaturated soil/saturated soil interface and/or the freezing interface. This difficulty on what is actually reflected is illustrated by the difference in the modeling and the experimental radargram. As presented, the M-GPR system, is fully autonomous and can monitor during 2 month at 30 minutes intervals with a $12\,V$ $60Ah$ car battery without any recharge. The field deployment is about to start having been delayed due to international tensions. That being said coupled with a set of temperature and





volumetric water content, the method holds a strong potential for bringing valuable thermo-hydro-dynamical information and refine the various processes taking place during melting and freezing.



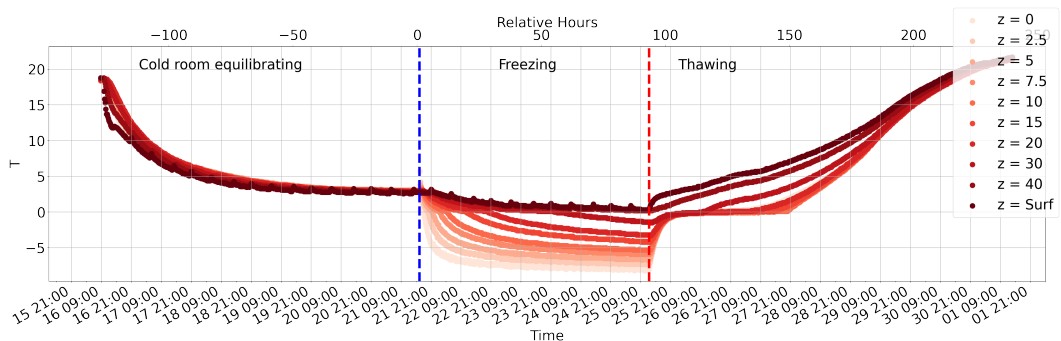

(a) Temperature monitoring at different height in the tank

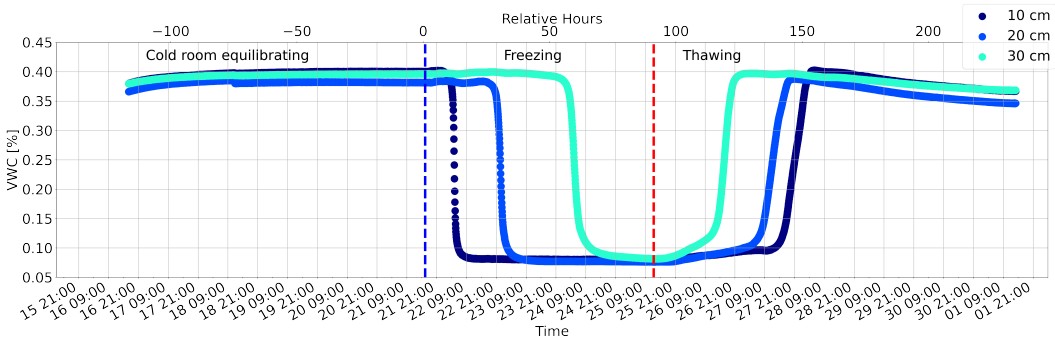

(b) Volumetric liquid water content monitoring

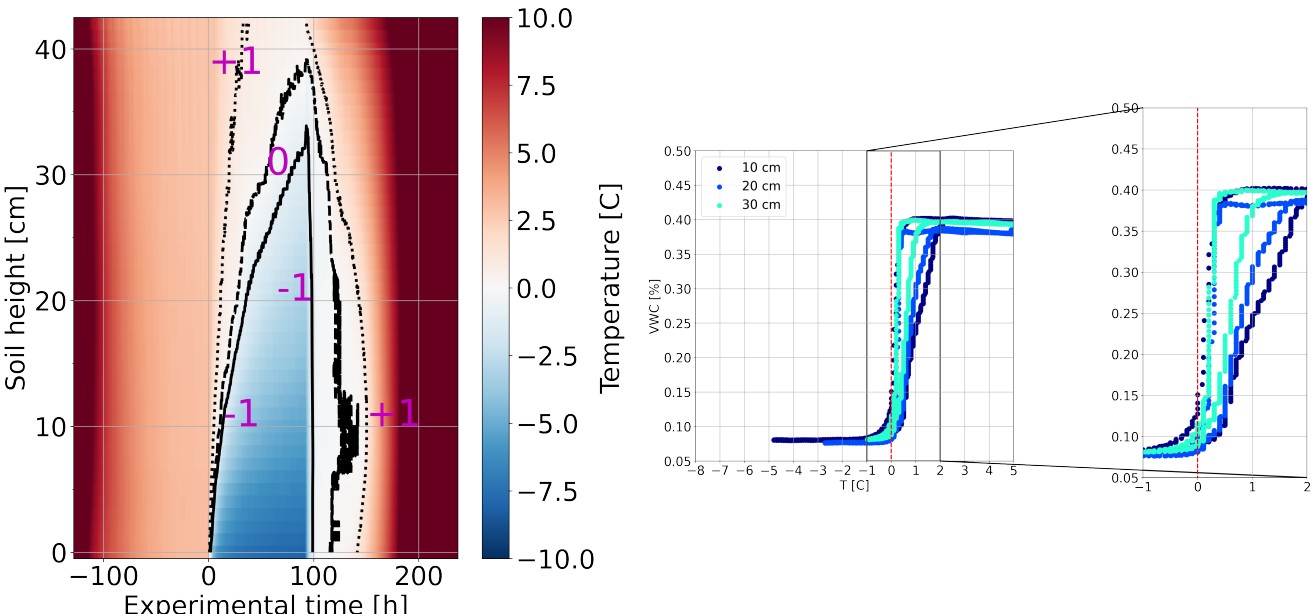

(c) Temperature monitoring depth profiles built from a)   (d) Soil Freezing Curve, each color being associated with a sensor height

**Figure 2.** Temperature and volumetric water content monitoring of the experiment : a) Thermal monitoring, b) volumetric water content monitoring, c) reconstructed 2D temperature column, d) Soil Freezing Curves.

(a) Simulated radargram (after median trace removel)

(b) Dielectric permittivity profiles

(c) Experimental radargram

(d) 0°C isotherm height as a function of time difference between A and C reflections

**Figure 3.** a) Modelled radargram with b) permittivity distributions and c) experimental radargram displayed with normalized amplitude. Labelled (A), (B) and (C) reflections correspond to the permafrost base, bottom and top of the transition zone. Dots in (a) are computed TWT using permittivity distributions in (b) where B and C positions are indicated for the $t = 0$. TWT stands for Two-Way travel Time. c) $\text{TWT}_C$ and $\text{TWT}_A$ difference as a function of isotherm-0°C height.



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

project INSU.

Critical reviews by Adam Booth and an anonymous reviewer greatly enhanced the presentation of this manuscript.

This article is dedicated to Christophe Grenier.