# Peer review of "Brief Communication: Monitoring active layer dynamic using a lightweight nimble Ground-Penetrating Radar system. A laboratory analog test case."

_The Cryosphere, 2022_

## Author Comment (AC1)

**Brief Communication : Monitoring active layer dynamic using a lightweight nimble Ground-Penetrating Radar system. A laboratory analog test case. Answer to reviewers**

Emmanuel Léger[1], Albane Saintenoy[1], Mohammed Serhir[3], François Costard[1], and Christophe Grenier[2,†]

[1]Université Paris-Saclay, CNRS, GEOPS, 91405, Orsay, France

[2,†]Laboratoire des Sciences du Climat et de l'Environnement, Université Paris-Saclay,IPSL/LSCE, UMR 8212 CNRS-CEA-UVSQ,Orme des Merisiers, Gif-sur-Yvette Cedex,France. Deceased

[3]Université Paris-Saclay, CentraleSupélec, CNRS, Laboratoire de Génie Electrique et Electronique de Paris, 91192, Gif-sur-Yvette, France

Dear reviewers RC1,

Thanks for your review. Please find below our answers to your comments .

**1 Originality (novelty): 2**

5 *In this contribution, the authors present a laboratory test using ground-penetrating radar to monitor the active layer dynamics. From the reviewer's understanding, the novelty of this contribution is that the authors used thermal and volumetric water content sensors as reference sensors in the experiment to understand the freezing/thawing process better.*

The aim of the study is to present a novel low-cost monitoring GPR system prior to field deployment. We slightly changed
10 the abstract by stating that the aim of the paper was to test a low-cost monitoring GPR, L.5 " The correspondence between the frozen front electromagnetic reflection and temperature allowed to test the ability of the system to closely monitor the frozen front/bottom of the active layer reflection.

We emphasize the aim of the study, as well, at the end of the introduction :

"Here we present a novel combination of low-cost/-energy nimble GPR mono-static ground-coupled antenna with a reflec-
15 tometer in conjunction with a small array of thermal and volumetric water content sensors for monitoring active layer freezing and thawing during laboratory experiments. The study is thought as a first test case on a active layer laboratory analog before near future field deployments."

**2 Scientific quality (rigour): 3**

*The experiment is well constructed under laboratory conditions. Section 2.1 is well written, except it is unclear whether three or four volumetric water content sensors are used.*

We clarified that we used three sensors for VWC measurements. L 48 "Complementing these thermal measurements, 3 volumetric water content sensors (Decagon Terros 12) were set diametrically opposed to the thermistors string at 0, 10, 20 and 30 cm from the bottom of the sand layer". We modified as well the Figure 1-a)

*The reviewer doubts if bedding the bowtie antenna by 30° can help to focus the energy.*

We removed the sentence, since we are not presenting multiple antenna tests in this present study.

*Equation (3) does not match its description (Line 91). The reviewer cannot understand why neither $\epsilon_w$ nor $\theta_i$ is given in equation (3). Further, in Section 3.2, the reviewer cannot understand how the dielectric permittivity distribution is derived from equation (3).*

It has been corrected, this was, as you pointed out as well, typos. We added as well, Line 91 : "A transition zone was assumed between these two types of media (frozen/ unfrozen) were coexistence between frozen and thawed media was derived using the measured freezing curve without considering any change in porosity through the freezing process."

**3 Significance (impact): 3**

*Since the main result is poorly presented, the reviewer cannot judge the significance of the contribution.*

We tried to clarified the result presentation. Figures 1, 2 and 3 have been changed according to your remarks and better legend have been written. Paragraph 3.2 have been edited and we simplified Figure 3. As the paper has to be kept as short as possible, we were not mentioning all necessary information as you pointed out, we hope it is now clearer and you will be able to judge.

**4 Presentation quality: 4**

*Poor writing style with many typos. The notations are not consistent, for example, S11 and zero-isotherm.*

Has been changed according to your remarks.

*Mistakes, e.g., line 103: "the depth of the zero-isotherm reaching the ground surface after 125h from the start..." does not match figure 2c).*

Has been changed according to your remark.

*Line 145: "efficiency ...is...less than a centimeter". What does this mean?*

has been change to L 155 :" For the laboratory conditions encountered in the study, the M-GPR method was able to monitor a moving interface with an uncertainty being less than a centimeter [...]"

*The quality of the figures is poor. Every figure has another font size.*
We tried to correct this, in this new version.

*The axis label of Figure 3(b) is even half covered. The reviewer cannot understand Figure 3(b). If we look at the t=0 curve, does it mean at the beginning of the experiment, the height of the sand is only about 0.1 m?*
We modified the figure 3b to have the axis labels visible. The figure 3b represents relative permittivity profiles at different simulated times, corresponding to a freezing front going upward. The frozen saturated sand has a relative permitivity of about 4, while the saturated sand reaches 23. As the frozen front is moving upward the permittivity is dropping from 23 to 4, following a sigmoidal shape (Soif Freezing Curve). This is the same kind of profile you would obtain during a capillary rise.

*Figure 3(d) should be an important result of the contribution. However, it is very poorly described. What are the black dots in the plot? Why are they not been used for calculating the linear regression?*

The figure has been re-drawn, the black dots were corresponding to the beginning of the experiment and thawing phase. We now keep only the freezing phase points. We plot the 0-degree isotherm height as a function of the two-way travel times between the bottom of the sand and the freezing front. As such, we obtain the time needed for the EM wave to propagate thought the frozen media and then its bulk permittivity. We changed and added a sentence on Line 143-145 : "[...] gives the time needed for the electromagnetic wave to propagate thought the frozen media as a function of its thickness. Figure 2-d) shows that the points (TWTs as a fonction of frozen front height) align in a linear relationship inversely [...]". We do hope and do believe it will help you understand.

Thanks for your relevant comments,
Best regards
Emmanuel Léger

---

## Author Comment (AC2)

**Brief Communication : Monitoring active layer dynamic using a lightweight nimble Ground-Penetrating Radar system. A laboratory analog test case. Answer to reviewers**

Emmanuel Léger[1], Albane Saintenoy[1], Mohammed Serhir[3], François Costard[1], and Christophe Grenier[2,†]

[1]Université Paris-Saclay, CNRS, GEOPS, 91405, Orsay, France

[2,†]Laboratoire des Sciences du Climat et de l'Environnement, Université Paris-Saclay,IPSL/LSCE, UMR 8212 CNRS-CEA-UVSQ,Orme des Merisiers, Gif-sur-Yvette Cedex,France. Deceased

[3]Université Paris-Saclay, CentraleSupélec, CNRS, Laboratoire de Génie Electrique et Electronique de Paris, 91192, Gif-sur-Yvette, France

Dear reviewers RC1,

Thanks for your review. Overall we edited consequently the paper in term of presentation. Hope it will suit you better. Please find below our answers to your comments .

**1 Originality (novelty): 2**

*In this contribution, the authors present a laboratory test using ground-penetrating radar to monitor the active layer dynamics. From the reviewer's understanding, the novelty of this contribution is that the authors used thermal and volumetric water content sensors as reference sensors in the experiment to understand the freezing/thawing process better.*

The aim of the study is to present a novel low-cost monitoring GPR system prior to field deployment. We slightly changed the abstract by stating that the aim of the paper was to test a low-cost monitoring GPR, L.5 " The correspondence between the frozen front electromagnetic reflection and temperature allowed to test the ability of the system to closely monitor the frozen front/bottom of the active layer reflection.

We emphasize the aim of the study, as well, at the end of the introduction :

"Here we present a novel combination of low-cost/-energy nimble GPR mono-static ground-coupled antenna with a reflectometer in conjunction with a small array of thermal and volumetric water content sensors for monitoring active layer freezing and thawing during laboratory experiments. The study is thought as a first test case on a active layer laboratory analog before near future field deployments."

We changed the order of the materials and methods parts, starting first by the M-GPR system. Part 2.1 is now the GPR system (L.25)

**2 Scientific quality (rigour): 3**

*The experiment is well constructed under laboratory conditions. Section 2.1 is well written, except it is unclear whether three or four volumetric water content sensors are used.*

We clarified that we used three sensors for VWC measurements. L 77 "Complementing these thermal measurements, 3 volumetric water content sensors (Decagon Terros 12) were set diametrically opposed to the thermistors string at 0, 10, 20 and 30 cm from the bottom of the sand layer". We modified as well the Figure 1-a) and the associated legend.

*The reviewer doubts if bedding the bowtie antenna by 30° can help to focus the energy.*

We removed the sentence, since we are not presenting multiple antenna tests in this present study.

*Equation (3) does not match its description (Line 91). The reviewer cannot understand why neither $\epsilon_w$ nor $\theta_i$ is given in equation (3). Further, in Section 3.2, the reviewer cannot understand how the dielectric permittivity distribution is derived from equation (3).*

It has been corrected, this was, as you pointed out as well, typos. We added as well, Line 91 : " A transition zone was assumed between these two types of media (frozen/ unfrozen) were coexistence between frozen and thawed media was derived using an empirical freezing curve (the sinus of the ice content percentage in pores decreasing linearly from 100 to 0) without considering any change in porosity through the freezing process."

**3 Significance (impact): 3**

*Since the main result is poorly presented, the reviewer cannot judge the significance of the contribution.*

We tried to clarified the result presentation. Figures 1, 2 and 3 have been changed according to your remarks and better legend have been written. Paragraph 3.2 have been edited and we simplified Figure 3. As the paper has to be kept as short as possible, we were not mentioning all necessary information as you pointed out, we hope it is now clearer and you will be able to judge.

**4 Presentation quality: 4**

*Poor writing style with many typos. The notations are not consistent, for example, S11 and zero-isotherm.*

Has been changed according to your remarks. We edited a lot of typos, errors and glitches.

*Mistakes, e.g., line 103: "the depth of the zero-isotherm reaching the ground surface after 125h from the start..." does not match figure 2c).*

Has been changed according to your remark.

*Line 145: "efficiency ...is...less than a centimeter". What does this mean?*

has been change to L 155 :" For the laboratory conditions encountered in the study, the M-GPR method was able to monitor a moving interface with an uncertainty being less than a centimeter [...]"

*The quality of the figures is poor. Every figure has another font size.*
We tried to correct this, in this new version.

*The axis label of Figure 3(b) is even half covered. The reviewer cannot understand Figure 3(b). If we look at the t=0 curve, does it mean at the beginning of the experiment, the height of the sand is only about 0.1 m?*
We modified the figure 3b to have the axis labels visible. The figure 3b represents relative permittivity profiles at different simulated times, corresponding to a freezing front going upward. The frozen saturated sand has a relative permitivity of about 4, while the saturated sand reaches 23. As the frozen front is moving upward the permittivity is dropping from 23 to 4, following a sigmoidal shape (Soif Freezing Curve). This is the same kind of profile you would obtain during a capillary rise.

*Figure 3(d) should be an important result of the contribution. However, it is very poorly described. What are the black dots in the plot? Why are they not been used for calculating the linear regression?*

The figure has been re-drawn, the black dots were corresponding to the beginning of the experiment and thawing phase. We now keep only the freezing phase points. We plot the 0-degree isotherm height as a function of the two-way travel times between the bottom of the sand and the freezing front. As such, we obtain the time needed for the EM wave to propagate thought the frozen media and then its bulk permittivity. We changed and added a sentence on Line 143-145 : "[...] gives the time needed for the electromagnetic wave to propagate thought the frozen media as a function of its thickness. Figure 2-d) shows that the points (TWTs as a fonction of frozen front height) align in a linear relationship inversely [...]". We do hope and do believe it will help you understand.

Thanks for your relevant comments,
Best regards
Emmanuel Léger

---

## Author Comment (AC3)

**Brief Communication : Monitoring active layer dynamic using a lightweight nimble Ground-Penetrating Radar system. A laboratory analog test case. Answer to reviewers**

Emmanuel Léger[1], Albane Saintenoy[1], Mohammed Serhir[3], François Costard[1], and Christophe Grenier[2,†]

[1]Université Paris-Saclay, CNRS, GEOPS, 91405, Orsay, France

[2,†]Laboratoire des Sciences du Climat et de l'Environnement, Université Paris-Saclay,IPSL/LSCE, UMR 8212 CNRS-CEA-UVSQ,Orme des Merisiers, Gif-sur-Yvette Cedex,France. Deceased

[3]Université Paris-Saclay, CentraleSupélec, CNRS, Laboratoire de Génie Electrique et Electronique de Paris, 91192, Gif-sur-Yvette, France

Dear reviewer RC2,

Thanks for your review and your useful comments. Overall we edited consequently the paper in terms of presentation and rephrased most of the results. We added additional materials to answer one of your concern. Please find below our answers to your comments .

5

*[...]However, only one experimental dataset has been collected which shows some interesting features from the freezing and thawing cycles, but there is no indication of repeatability and sensitivity of experimental parameters. The numerical model is really too simplified and basic to be of much value. It would have been interesting and much more inciteful to include the GPR antenna and dispersive effects from the water (particularly during freezing/thawing) in the numerical model.*

10    We performed 9 experiments in total, involving each time, the freezing and warming cycle of the cold chamber/room, the partial or full saturation of the $approx$ 400 kg of dry sand and its correct compaction. The freezing and thawing phase is taking about 10 days with the thermal equilibrium of the chamber and the bottom freezing procedure with the cryostat. We have a total of 6 good experiments, defined here as non perturbed with power supply failure, leakage and container breaking because of the temperature. We insist once again that the aim of the paper is to present the capability of the system to monitor changes

15 in permafrost table height and/or thawed layer movement. This is why we privileged describing and showing one experiment only, keeping the message short (only 4 pages for a brief communication) and simple. In order to illustrate our answer to your very good comment we added : L.106-107 "We present in detail the results of one experiment while 4 others are available in supplementary materials."

   Concerning the numerical modeling, we re-explained it and added 2 other figures to strengthen the message. However we agree

20 with you in the sense that the modeling is very rough, once again aimed to understand which reflection is coming from where. This is especially difficult to understand the difference between B and C reflection origin and our "simple" modeling allows us to give a first hypothesis. The next step would be to model entirely the thermo-hydrodynamical process and then couple it

with GPR-codes. This is out of the scope of this study since, once again we target a paper on the prototype proof of concept. We added 2 sentences :

25   L 137-138 "We use this simple modeling to interpret the experimental radargram.";

L. 160-163 "The modeling presented in this study is very simple, but was aimed in understanding from where the reflections observed in the experimental radargram where coming from. A coupled thermo-hydro-EM modeling is currently in development in order to fully simulate the physical phenomena and use the M-GPR data to infer SFCs."

30   *[...]The presentation of the manuscript needs to be improved - there is a lack of consistency of style across the figures and the text contains grammatical errors. The features of the measured GPR data (Figure 3c) could be better highlighted and described in the text.*

We answered these issues with our RC1 comments. We redesigned all the figures and re-explained in details Figures 2 and 3. We homogenized the fonts and swiped part for clarity. We do hope it will suit you better.

35

Thanks for your relevant comments,

Best regards

Emmanuel Léger